

**Estimation of Metabolic Dynamics of Restored Seagrass Meadows in a Southeast Asia Islet:**
**Insights from Ex Situ Benthic Incubation**
Mariche B. Natividad[123*], Jian-Jhih Chen[45*], Hsin-Yu Chou[1], Lan-Feng Fan[1], Yi-Le Shen[6], Wen-Chen
Chou[178]
[1]Institute of Marine Environment and Ecology, National Taiwan Ocean University, Taiwan
[2]Doctoral Degree Program on Ocean Resources and Environmental Changes, College of Ocean Science
and Resources, National Taiwan Ocean University, Taiwan
[3]Ecosystems Research and Development Bureau, Laguna, Philippines
[4]Department of Marine Environmental Engineering, National Kaohsiung University of Science and
Technology, Taiwan
[5]Department of Oceanography, National Sun Yat-Sen University, Taiwan
[6] Penghu Fisheries Biology Research Center, FRI, MOA, Taiwan
[7]Center of Excellence for the Oceans, National Taiwan Ocean University, Keelung, Taiwan
[8]Institute of Marine Biology, National Dong Hwa University, Pingtung, Taiwan
*Correspondence to*: Wen-Chen Chou ([wcchou@mail.ntou.edu.tw](mailto:wcchou@mail.ntou.edu.tw))
* These authors contribute equally.
**Abstract.** Seagrass meadows are vital carbon sinks, but their function is threatened by rapid decline,
driving restoration efforts to enhance coastal recovery and carbon removal. The capacity of these restored
seagrass as carbon sources or sinks depends largely on organic carbon metabolism and carbonate
dynamics. In this study, we employed ex situ core incubation to investigate the metabolic rates of
replanted seagrasses (SG), including gross primary productivity (GPP), community respiration (R), net
ecosystem metabolism (NEM), and net ecosystem calcification (NEC) in SG and surrounding bare
sediments (BS). SG exhibited higher GPP ($26.0 \pm 1.0$ mmol $O_2$ m$^{-2}$ h$^{-1}$ *vs* $0.7 \pm 0.1$ mmol $O_2$ m$^{-2}$ h$^{-1}$) and
NEM ($208.2 \pm 6.3$ mmol $O_2$ m$^{-2}$ d$^{-1}$ *vs* $20.1 \pm 2.8$ mmol $O_2$ m$^{-2}$ d$^{-1}$) than BS, indicating their potential as
carbon sinks by shifting benthic metabolism towards a more autotrophic state. In contrast, SG showed
higher daytime carbonate production and nighttime carbonate dissolution, which could offset each other,
resulting in no significant difference in NEC between SG and BS. In summary, our results found that the
SG exhibited significantly higher NEM compared to BS, while no significant difference was found for
NEC. Consequently, the net effect on the carbon uptake capacity of the restored seagrass is likely





increased, primarily due to the higher NEM. Our findings highlight the ecological significance of seagrass
restoration in mitigating climate change through carbon removal. Ex situ core incubation method allows
for the simultaneous measurement of organic and inorganic carbon metabolism. While ex situ core
incubation enhances feasibility, in situ assessments are still necessary to validate the results and ensure a
comprehensive understanding of seagrass ecosystem dynamics.

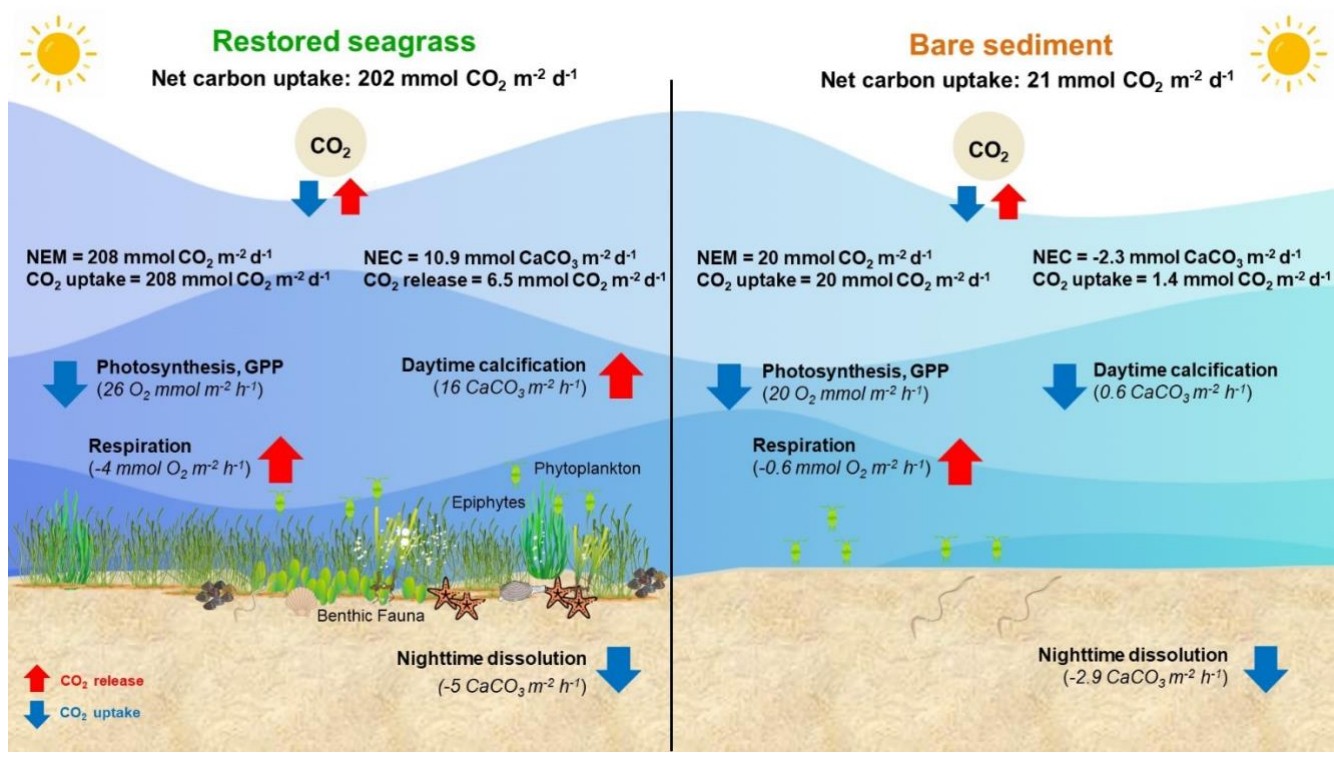


**Graphical abstract: Illustration of carbon uptake from organic carbon metabolism (GPP-gross**
**primary productivity, R-respiration, NEM-net ecosystem metabolism) and carbonate dynamics**
**(daytime calcification, nighttime dissolution, and NEC-net ecosystem calcification) in restored**
**seagrass and bare sediment. Net Ecosystem Metabolism (NEM).**



## 1 Introduction

**1 Introduction**
Seagrass meadows, comprising over 72 species, occupy just 0.1% of the ocean's surface, yet they are
highly productive and ecologically significant ecosystems in the marine environments (Fourqurean et al,
2012; Short et al., 2011). These meadows play essential roles in nutrient and carbon cycling and serve as
key habitats for many marine species (Duarte et al., 2010; Fourqurean et al., 2012). Due to their relatively
complex structure, seagrass meadows capture and retain organic carbon ($C_{org}$) in the sediment, making
them one of the major carbon reservoirs globally (Duarte et al., 2005; Mcleod et al., 2011). Previous
estimates suggest that seagrasses account for approximately 15% of the total global carbon sequestered
in benthic sediments (Duarte et al., 2013), with burial rates 35 times that of tropical rainforests (Mcleod
et al., 2011).

In spite of their ecological significance, seagrass meadows have experienced a global decline, driven
primarily by human-induced activities such as coastal development, eutrophication, and deteriorating
water quality (Orth et al., 2006; Waycott et al., 2009). Since 1980, the global coverage of seagrass has
decreased by 110 km² annually, with the rate of decline increasing (Waycott et al., 2009). The loss is
frequently associated with increased water column turbidity and epiphytic shading, which reduce the light
for seagrass photosynthesis, leading to meadow degradation (Campbell et al., 2003; Orth et al., 2006).
Degradation also diminishes their capacity to modify local pH and influence the dynamics of dissolved
oxygen (DO) and dissolved inorganic carbon (DIC) (Hendricks et al., 2014). Moreover, the continued
loss of seagrass ecosystems raises concerns that vast amounts of previously sequestered carbon could be
released back in the atmosphere, converting seagrasses from carbon sinks to carbon sources and
intensifying global climate change (Macreadie et al., 2013). The ongoing decline could potentially release
up to 299 Tg of carbon annually, contributing roughly 10% of $CO_2$ emissions associated with
anthropogenic land-use changes (Fourqurean et al., 2012).

In response to these challenges, seagrass restoration has emerged as a critical strategy to mitigate
environmental degradation, enhance coastal resilience, and address global climate change (Juska and Berg



et al., 2022). Protecting and restoring seagrass meadows aligns with international goals like the Paris
Agreement, as these ecosystems offer significant potential for long-term carbon storage and climate
regulation (Fourqurean et al., 2012). However, despite growing restoration efforts, there remains limited
understanding of their success, particularly regarding benthic metabolism and carbon dynamics
(Kindeberg et al., 2024). While studies from temperate regions, such as the *Zostera marina* restoration in
the Virginia Coast (Rheuban et al., 2014), have provided valuable insights, data from tropical regions —
including Southeast Asia, a global hotspot for seagrass diversity — remain scarce (Duarte et al., 2010;
Ward et al., 2022; Chou et al., 2023). It represents a critical gap in our knowledge of the impact of
restoration efforts on carbon removal and ocean acidification mitigation.

Although there is increasing consensus on the potential of "Blue Carbon" storage in seagrass meadows
as a climate change mitigation strategy, the biogeochemical cycling within these ecosystems is complex.
Several processes, including ecosystem calcification, anaerobic metabolism, and bioturbation, can
counteract net organic carbon (OC) sequestration (Van Dam et al., 2021). These processes regulate local
DIC and total alkalinity (TA) budgets, adding complexity to accurately quantifying carbon sequestration
(Kindeberg et al., 2024). Overlooking these processes can result in significant overestimates of local
carbon sequestration rates and misinterpretations of the role seagrass meadows play in mitigating climate
change, potentially leading to inaccurate assessments of their carbon sink capacity (Johansen et al., 2023;
Chen et al., 2024; Fan et al., 2024).

Several methodologies were developed to quantify benthic metabolism, which is a crucial component of
biogeochemical cycling, including photosynthesis-irradiance curve (Kraemer and Alberte, 1993), the
open water $O_2$ mass balance approach (Odum, 1956; Chou et al., 2023), and aquatic eddy covariance
(Berg et al., 2022; Juska and Berg, 2022). While these methods provide important data, they might
overlook the complexities of bioturbation, remineralization, and carbonate dynamics (Olive et al., 2016;
Ward et al., 2022; Juska and Berg, 2022). In this study, we aim to address these knowledge gaps by
quantifying organic carbon metabolism (net ecosystem metabolism, NEM) and carbonate dynamics (i.e.,



net ecosystem calcification, NEC) in restored seagrass meadows (SG) and adjacent bare sediment (BS)
habitats on a Southeast Asia islet, using an innovative ex situ benthic incubation.

## 2 Materials and Methods

### 2.1 Study site

The Penghu Islands, located in the southern part of Taiwan Strait (Fig. 1), host a range of seagrass species.
Notably, four species have been reported: *Halophila ovalis*, *Halodule pinifolia*, *Halodule uninervis*, and
*Zostera japonica* (Yang et al., 2002). The sampling location (23° 38' 18.38" N and 119° 33' 46.48" E) is
a restoration meadow dominated by *H. uninervis* and *H. ovalis*. This restoration site encompasses
approximately 3 hectares as per Coral Alen Atlas, with seagrass percent cover varying from 20% to 90%.
These seagrasses are subtidal, with water depths ranging from 1.7 meters to 4.4 meters. The substrate in
this area is composed of carbonate sand. The area supports a diverse community of bivalves (e.g., *Pinna*
sp.), gastropods, echinoderms, and various fish species, all of which were observed during the sampling.



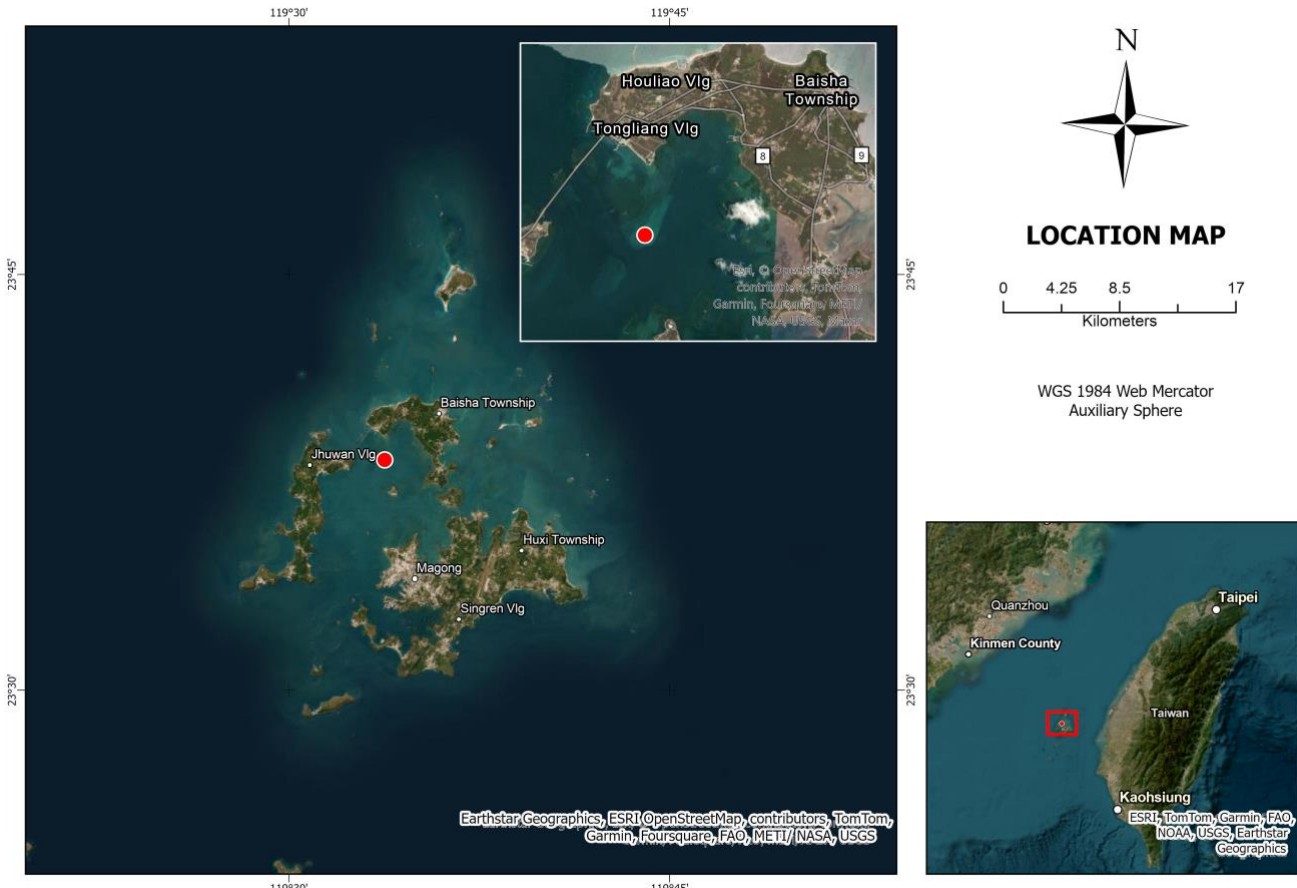


**Figure 1: Location map of sampling stations in restored seagrass in Penghu Island, Taiwan (Map created in ArcGIS Pro. Source: Earthstar Geographics, ESRI OpenStreetMap, Contributors, TomTom, Garmin, Foursquare, FAO, METI/NASA, USGS, NOAA).**

**2.2 Ex situ core incubation system**

The ex situ benthic core methodology used in this study was adapted from Chen et al. (2019) (Fig. 2). This approach has been widely employed in various studies to assess nutrient concentrations and benthic metabolism in coastal ecosystems and estuaries (Eyre & Ferguson, 2005; Maher & Eyre, 2011). Typically, the ex situ core incubation involves 150-L treatment tanks containing aerated water. Each tank can accommodate 10 plexiglass cores made of polycarbonate material, 10 cm in diameter and 50 cm in height. The tanks were equipped with magnetic stir bars driven by a centrally located rotating motor fitted with



a magnet. The core has a plexiglass lid which contains two ports, one for probe insertion (Eyre &
Ferguson, 2005).

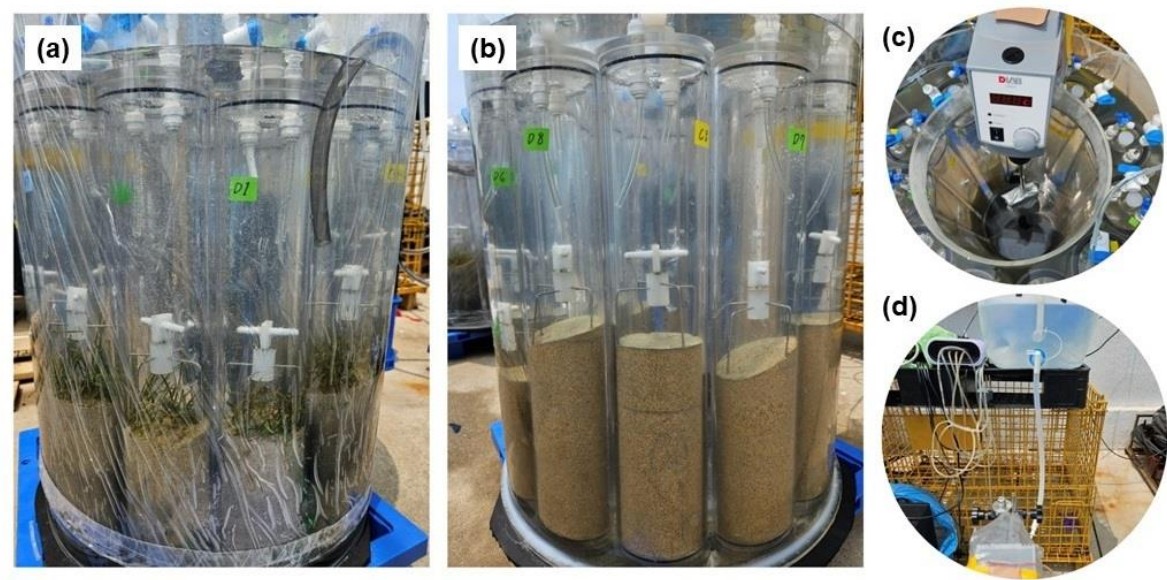


**Figure 2: Ex situ benthic chamber setup for measuring metabolic rates and carbonate dynamics in**
**seagrass meadows and bare sediment. The chambers contain seagrass samples (a), while the**
**chambers contain bare sediment (b). Insets show close-ups of the central rotating motor with a**
**magnet setup for water circulation (c), and the setup for continuous seawater supply (d).**
**2.3 Sediment core collection and pre-incubation**
The incubation was conducted on April 12-13, 2024. Twenty intact sediment cores, comprising both
seagrass and bare sediment, were collected on-site using the plexiglass tubes. The cores were inserted
about 20 cm into the sediment, keeping approximately 1.9 liters of water. Each core was sealed with a
gas–tight plexiglass plate at the bottom. The samples were brought back to the incubation site within two
hours of collection and allowed to settle for 24 hours. Additionally, 150 liters of water were collected on-
site for continuous supply during the experiment.



At the incubation site, the cores were uncovered and placed in 150–liter tanks filled with aerated seawater.
They were kept at in situ temperature, exposed to natural sunlight, and continuously recirculated. The
stirring rate was controlled to prevent sediment resuspension (Ferguson et al., 2004). The cores underwent
a 24–hour pre–incubation period to promote stable sediment profiles. The seagrass composition within
the collected cores for ex situ core incubation was dominated by *H. uninervis* and *H. ovalis*. The shoot
count of *H. uninervis* ranged from 20 to 40 shoots per 0.008 m², while *H. ovalis* ranged from 2 to 20
shoots per 0.008 m².
**2.4 Sample collection and analysis**
Following pre-incubation, the cores was tightly closed using a plexiglass lid. Temperature, salinity, and
pH were determined using a YSI ProDSS Multiparameter water quality checker, while DO (mg $l^{-1}$) was
measured with a thermo DO probe. Both probes were calibrated with calibration standards. Measurements
were taken at midnight (24:00 h) with 2-hour intervals and ended at noon. Photosynthetically active
radiation (PAR) levels were measured using SQ-420X Smart Quantum Sensor positioned atop the
incubation tank.

After measurements, three 150 ml seawater samples were collected separately from the SG and BS cores
using a syringe for DIC and pH analysis. The water samples were processed with 60 μL $HgCl_2$ solution
to stop any biological activity. DIC analysis was performed using a non–dispersive infrared method with
a DIC analyzer (AS-C3, Apollo SciTech Inc.), following the approach of Dickson et al. (2007) and our
past studies (Chou et al., 2018; 2021; Fan et al., 2024). For each DIC run, we used certified reference
material (Batch no. 206) sourced from A. G. Dickson at Scripps Institution of Oceanography to check for
drift and systematic bias. pH values were measured spectrophotometrically in total scale at 25 °C
following Clayton and Byrne (1993). Data from DIC and pH, along with actual temperature and salinity,
were used to calculate the TA, partial pressure of $CO_2$ ($p$$CO_2$), and aragonite saturation state ($\Omega_{Ar}$) using
the Excel macro CO2SYS version 2.1 (Pelletier et al. 2011). The dissociation constants for carbonic acid
applied in these calculations were obtained from Mehrbach et al. (1973) and subsequently refined by
Dickson and Millero (1987).



## 2.5 Benthic flux rate calculations

Areal rates of R, GPP, NPP, and NEM were calculated based on changes in DO concentrations, following equation 1 (Eyre et al. 2011). Respiration rates were determined from concentration data collected during the initial dark period (midnight to dawn) (eq. 2). NPP was calculated based on light $O_2$ flux measurements from dawn to noon (eq. 3). Hourly GPP rates were computed as the difference between R and NPP rates (eq. 4). NEM was calculated using equation 5. Positive values indicate autotrophic, while negative values represent heterotrophic.

$$F = [(C_{t1} - C_{t0}) \times V/A]/T] \qquad \text{(eq. 1)}$$

Where F = flux rate ($\mu$mol m$^{-2}$ h$^{-1}$), $C_{t0}$ and $C_{t1}$ = concentration in the overlying water at the start and end of the time period ($\mu$mol l$^{-1}$), respectively, V = volume of overlying water in the core (l), A = surface area in the sediment core (m$^2$), and T = incubation period (h).

$$R = \text{dark } O_2 \text{ flux (negative)} \qquad \text{(eq. 2)}$$

$$NPP = \text{light } O_2 \text{ flux (positive)} \qquad \text{(eq. 3)}$$

$$GPP = NPP - R \qquad \text{(eq. 4)}$$

$$NEM = (GPP \times 12) - (R \times 24 \text{ h} \times -1) \qquad \text{(eq. 5)}$$

NEC rates (mmol CaCO$_3$ m$^{-2}$ h$^{-1}$) were estimated from the change of total alkalinity, assuming these changes are only due to CaCO$_3$ precipitation and dissolution (eq. 6) (Roth et al., 2019; Van Dam et al., 2019):

$$NEC = -0.5 \frac{\Delta n\text{TA}}{\Delta t} \times hp \qquad \text{(eq. 6)}$$

Here, $\Delta n$TA = change in $n$TA ($n$TA = TA x SSS$_{average}$/SSS) over the $\Delta$t (time), h = Volume/Area, and $p$ = water density. The $-0.5$ scalar factor was applied to account for the stoichiometric relationship, where 2 moles of TA produce 1 mole of CaCO$_3$. Day and night incubations were conducted to obtain daily NEC fluxes. NEC is positive with TA consumption, indicating CaCO$_3$ precipitation, and negative with TA production, indicating CaCO$_3$ dissolution.



## 2.5 Statistical analysis

Independent sample T–tests were applied to compare metabolic rates (R, NPP, GPP, NEM, NEC) between SG and BS using SPSS v. 17. Data were subjected to a normality test before performing the analysis. Least–squares linear regression was employed to assess the correlation between changes in DO in the SG and BS. The Mann–Whitney U test was applied for carbonate chemistry analysis due to the non-normal distribution of data.

## 3 Results

### 3.1 Water quality and carbonate chemistry

Diurnal patterns of water quality and carbonate parameters for SG and BS during the two-day ex situ core incubation are illustrated in Figs. 3 and 4, respectively. The temperature in both treatments ranged from 22 to 29 °C, while salinity levels spanned from 35 to 36. These values were similar to in situ measurements obtained from the seagrass beds using a CTD profiler. During the daytime (6:00 AM to 12:30 PM), PAR levels ranged from 26 $\mu$mol m$^{-2}$ s$^{-1}$ to a peak of 1662 $\mu$mol m$^{-2}$ s$^{-1}$, with the highest intensities observed at midday. The average PAR measured 953 $\mu$mol m$^{-2}$ s$^{-1}$ on the first day of incubation, increasing slightly to 1026 $\mu$mol m$^{-2}$ s$^{-1}$ on the second day. DO saturation levels were more variable in SG than BS, with values ranging from 54% to 224% and 92% to 123%, respectively. DO saturation levels in both treatments followed a diel pattern, with lower nighttime and higher daytime values.

Both $n$DIC ($n$DIC = DIC x SSS$_{average}$/SSS) and pH$_T$ displayed greater diurnal fluctuations at SG compared to the BS. At SG, $n$DIC ranged from 1660 to 2118 $\mu$mol kg$^{-1}$ (mean ± SE: 1963 ± 41 $\mu$mol kg$^{-1}$), and followed a diel pattern. pH$_T$ ranged from 7.81 to 8.37 at SG (mean ± SE: 7.99 ± 0.05), following the opposite trend to $n$DIC, with values decreasing at night and increasing during the day. This daytime increase in pH$_T$ at SG indicated the potential role of seagrass in mitigating ocean acidification effects during daylight hours. At the BS site, these parameters were less variable, with $n$DIC values ranging from 1948 to 2029 $\mu$mol kg$^{-1}$ and pH$_T$ from 7.84 to 7.99, with mean values of 1993 ± 7 $\mu$mol kg$^{-1}$ and 7.93 ± 0.01, respectively. Similarly, the calculated $n$TA was also more fluctuating in SG than BS, with mean



values of $2243 \pm 6$ µmol kg⁻¹ and $2230 \pm 6$ µmol kg⁻¹, respectively. The calculated $p$CO₂ displayed a
broader range at SG (142 to 762 µatm; mean $\pm$ SE: $510 \pm 62$) compared to BS (450 to 699 µatm; mean $\pm$
SE: $524 \pm 22$), suggesting a more dynamic carbon cycling potentially driven by seagrass metabolic
activity. The mean $\Omega_{Ar}$ was higher in SG ($3.14 \pm 0.37$) compared to BS ($2.72 \pm 0.11$), indicating more
favorable conditions for calcification at the seagrass site. Mann–Whitney test on carbonate chemistry
revealed no significant distinction between SG and BS (pH$_T$ $p = 0.713$; $n$DIC $p = 0.419$; $n$TA $p = 0.679$;
$\Omega_{Ar}$ $p = 0.511$).

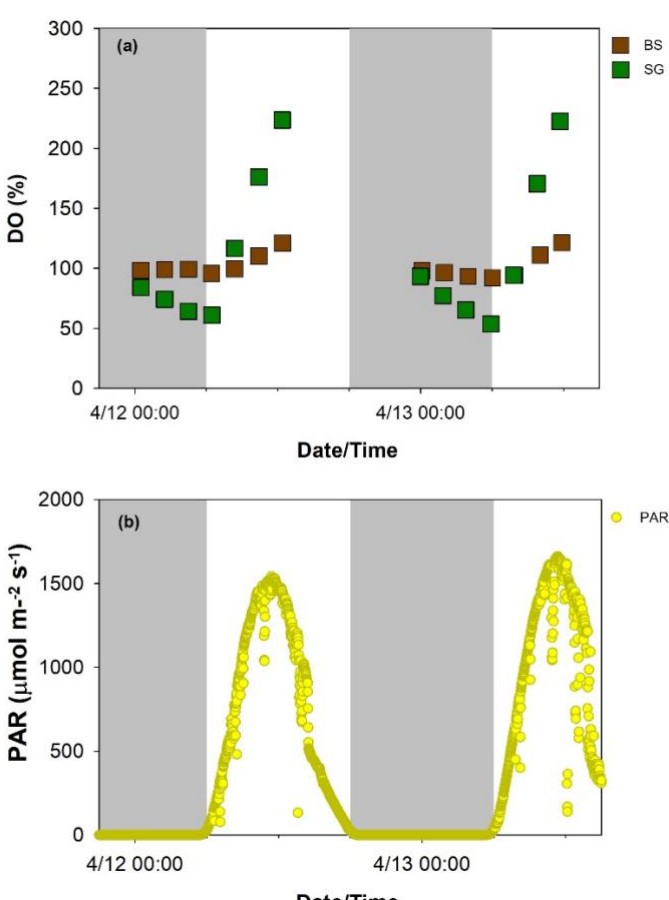


**Figure 3: Diurnal pattern of dissolved oxygen (DO, a) in replanted seagrass (SG, green square) and bare sediment (BS, brown square), and photosynthetically active radiation (PAR, b) during the two-day (April 12-13, 2024) incubation.**





Figure 4: Total scale pH (pH$_T$, a), normalized dissolved inorganic carbon ($n$DIC, b), normalized total alkalinity ($n$TA, c), partial pressure of carbon dioxide ($p$CO$_2$, d), and aragonite saturation state ($\Omega$A$_R$, e) in replanted seagrass (SG, green square) and bare sediment (BS, brown square) during the two-day (April 12-13, 2024) incubation. n=3, mean ± SE.



## 3.2 Respiration, gross primary production, and net ecosystem metabolism

Figure 5 illustrates the comparison of metabolic rates between SG and BS. The mean respiration rates in SG ($-4.3 \pm 0.3$ mmol $O_2$ $m^{-2}$ $h^{-1}$) were significantly higher than in BS ($-0.6 \pm 0.1$ mmol $O_2$ $m^{-2}$ $h^{-1}$), by approximately 8-fold difference ($p<0.01$). The mean GPP in SG was $26.0 \pm 1.0$ mmol $O_2$ $m^{-2}$ $h^{-1}$, which is 35-fold higher than in BS ($0.7 \pm 0.1$ mmol $O_2$ $m^{-2}$ $h^{-1}$) ($p<0.01$). GPP was always higher than R in both systems, with mean GPP/R ratios of 3.4 and 1.9 in SG and BS, respectively. For NEM, both systems displayed positive values, indicating net autotrophy, with SG being 10-fold higher ($208.2 \pm 6.3$ mmol $O_2$ $m^{-2}$ $d^{-1}$) compared to BS ($20.1 \pm 2.8$ mmol $O_2$ $m^{-2}$ $d^{-1}$) ($p<0.01$). Both R and GPP in SG and BS increased on the second day of incubation, while NEM showed a slight decrease, but these changes were not statistically significant.





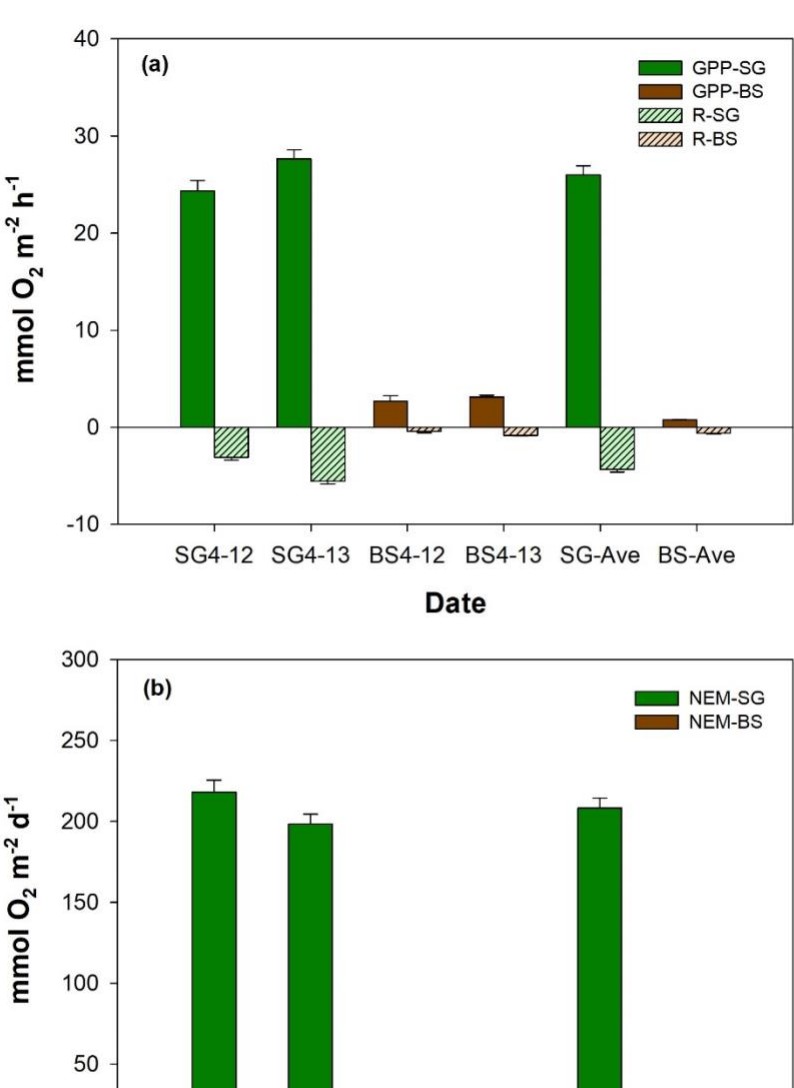

**Figure 5: Mean (± SE, standard error) values of (a) metabolic rates such as respiration (R), gross primary productivity (GPP), and (b) net ecosystem metabolism (NEM,) of restored seagrass (SG, green bars) and bare sediment (BS, brown bars) in Penghu during the two-day (April 12-13, 2024) incubation (n=9).**



## 3.2 Calcium carbonate precipitation, dissolution, and net ecosystem calcification

The NEC values over a diel cycle for SG and BS demonstrated differences in their overall carbonate dynamics (Fig. 6). Over the two-day incubation period, SG exhibited a net calcifying system with a mean positive daily NEC means ($10.9 \pm 5.1$ mmol $CaCO_3$ m$^{-2}$ d$^{-1}$), driven by daytime calcification ($16.1 \pm 3.7$ mmol $CaCO_3$ m$^{-2}$ h$^{-1}$) despite nighttime dissolution ($-5.2 \pm 3.9$ mmol $CaCO_3$ m$^{-2}$ h$^{-1}$). In contrast, BS supported a net-dissolving system with mean daily NEC ($-2.3 \pm 6.2$ mmol $CaCO_3$ m$^{-2}$ d$^{-1}$). Mean daytime calcification and nighttime dissolution were $0.6 \pm 6.4$ mmol $CaCO_3$ m$^{-2}$ h$^{-1}$ and $-3.0 \pm 0.9$ mmol $CaCO_3$ m$^{-2}$ h$^{-1}$, respectively. Both systems followed a general diurnal pattern, with positive NEC during the day (calcifying) and negative at night (dissolving).





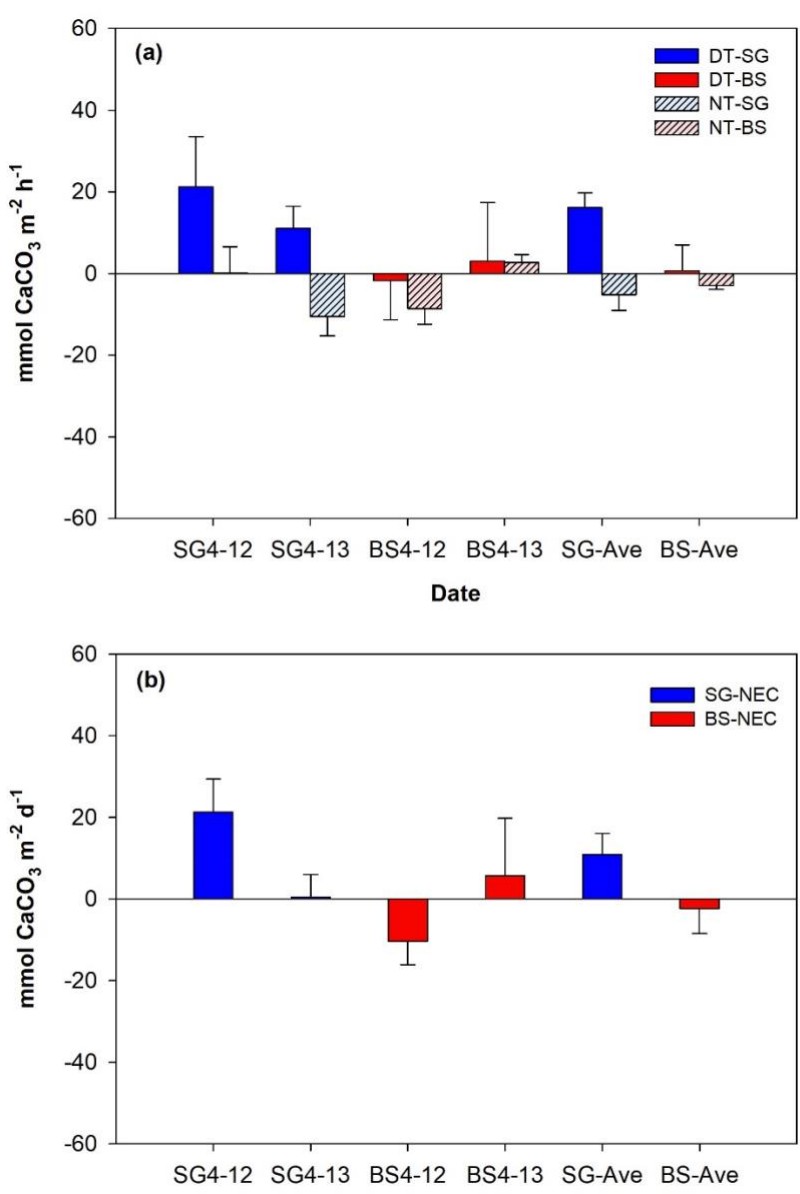

**Figure 6: Mean (± SE, standard error) values of daytime (DT) and nighttime (NT) calcification (a), and net ecosystems calcification (NEC, b) of restored seagrass (SG, blue bars) and bare sediment (BS, red bars) in Penghu during the two-day (April 12-13, 2024) incubation (n=9).**





## 4 Discussion

Seagrass meadows are widely recognized as an important blue carbon ecosystem with substantial potential to mitigate anthropogenic $CO_2$ emissions. Although research on seagrass ecosystems has grown in recent years, significant gaps remain in understanding their carbon dynamics. In particular, the balance of organic and inorganic carbon processes within these systems is not fully understood. Meanwhile, global seagrass coverage continues to decline, which has increased the urgency of restoration efforts (Waycott et al. 2009). Restoring seagrass meadows to enhance carbon sequestration has become increasingly important. Currently, most studies on restored seagrass meadows focus primarily on the burial of particulate organic carbon (Greiner et al. 2013), with far fewer exploring both organic metabolism and carbonate cycling in restored seagrass meadows. Here, we present the first dataset on carbon uptake through metabolic rates and calcification measurements in restored seagrass meadows within tropical regions.

### 4.1 Restoration of seagrass enhances metabolic rates

The metabolic rates estimated in present study were comparable to those recorded in other seagrass meadows (Table 1). Our GPP in SG was 24% and 37% higher than the tropical and global averages, respectively, but 38% lower than Dongsha Island, Taiwan (Chou et al., 2023). It is also comparable to measurements reported for *H. uninervis* in Tropical Australia (Table 1). Conversely, the R values estimated in this study were roughly half lower than the tropical and global averages (Duarte et al., 2010). Our NEM (214 mmol $O_2$ $m^{-2}$ $d^{-1}$) is within the range of previous estimates for tropical seagrass meadows (-477.28 to 484.20 mmol $O_2$ $m^{-2}$ $d^{-1}$) and global estimates (-477.28 to 531.63 mmol $O_2$ $m^{-2}$ $d^{-1}$). In addition to these global comparisons, our study reveals a clear distinction in metabolic rates (e.g. GPP, R, NEM) between SG and BS. The GPP and R in restored seagrass meadows were 35 and 7 times greater than in BS. The relatively higher metabolic rates in seagrass meadows compared to bare sediments have also been observed in other studies (Table 1). For instance, a two-year-old restored *Halodule wrightii* meadow demonstrated a 13-fold increase in NEM relative to bare sediment (Egea et al., 2023). Similarly, *Posidonia oceanica* exhibited a notable 70-fold increase in metabolic rates compared to bare sediment (Barron et al., 2006). Furthermore, *Zostera marina* exhibits net autotrophy while bare sediments are net





heterotrophy (Attard et al., 2019; Chen et al., 2019). Such patterns highlight the fundamental ecological functions restored seagrass meadows play relative to unvegetated/bare sediments. The increase in GPP reflects the enhanced carbon fixation capacity of seagrass meadows, while the elevated R indicates active organic matter decomposition and microbial respiration (Duarte and Krause-Jensen, 2017). According to Duarte et al. (2010), seagrass meadows generally become autotrophic (NEM > 0) when GPP is greater than 186 mmol $O_2$ $m^{-2}$ $d^{-1}$, shifting to heterotrophy (NEM < 0) at lower levels. Based on this threshold, our mean GPP for restored seagrass exceeded the value for autotrophy, resulting in a positive NEM which is consistent with their global assessment. The NEM observed in SG was 10 times higher than in BS, suggesting that SG sequesters significantly more carbon than BS. These findings highlight that seagrass restoration significantly boosts metabolic rates and enhances carbon cycling. Given the increasing loss of global seagrass cover, restoration not only boosts ecosystem productivity but also strengthens the ability of coastal systems to remove carbon, thereby contributing to climate change mitigation efforts.

**Table 1. Comparison of metabolic rates from global estimates. GPP and R values are expressed in mmol $O_2$ $m^{-2}$ $h^{-1}$ units, while NEM in mmol $O_2$ $m^{-2}$ $d^{-1}$.**

| Location | Method | Seagrass Community | GPP | R | NEM | References |
|---|---|---|---|---|---|---|
| Taiwan | Ex situ benthic chambers | Bare sediment | 0.74 ± 0.09 | 0.62 ± 0.09 | 20.10 ± 2.84 | This study |
| | | *H. uninervis, H. ovalis* | 25.99 ± 0.96 | 4.32 ± 0.26 | 208.21 ± 6.33 | |
| Taiwan | Open water mass balance | *Thalassia, Cymodocea* | 42.25 ± 14.42 | 20.71 ± 7.13 | 8 ± 61 | Chou et al., 2023 |
| Mexico | In situ benthic chambers | Bare sediment | 2.13 ± 0.58 | 0.73 ± 0.16 | 8.1 ± 10.9 | Egea et al., 2023 |
| | | 2-year *H. wrightii* | 13.76 ± 3.35 | 2.61 ± 0.40 | 102.4 ± 31.5 | |
| | | 4-year *H. wrightii* | 9.24 ± 2.34 | 1.60 ± 0.19 | 72.5 ± 27.9 | |
| | | 4-year *H. wrightii* | 9.34 ± 0.35 | 2.15 ± 0.25 | 60.7 ± 4.7 | |
| Sweden | Aquatic eddy covariance and benthic chambers | 3-year-old restored seagrass (*Z. marina*) | | | −5 to −15 | Kindeberg et al., 2024 |
| | | 7-year-old restored seagrass (*Z. marina*) | | | −21 | |
| Finland | | Bare sediment | 1.60 | 0.82 | -0.14 | |





|  |  |  |  |  |  |  |
|---|---|---|---|---|---|---|
|  | Aquatic eddy covariance | *Z. marina* | 3.74 | 1.71 | 4.17 | Attard et al., 2019 |
| Australia | Ex situ benthic | Bare sediment | 2.28 | 1.26 | -2.74 | Chen et al., 2019 |
|  |  | *Zostera* sp. | 6.94 | 2.74 | 7.12 |  |
|  |  | *Halophila* sp. | 2.05 | 1.60 | -13.70 |  |
| Tropical Australia | Combined methods | *H. uninervis* | 23.42 ± 3.67 | 9.63 ± 4.04 | 50 ± 53 | Duarte et al., 2010 |
| Tropical | Combined methods | All species | 21 ± 0.6 | 9 ± 0.6 | 24 ± 8 | Duarte et al., 2010 |
| Global | Combined methods | All species | 19 ± 0.5 | 8 ± 0.4 | 27 ± 6 | Duarte et al., 2010 |
| Spain | In situ benthic chambers | Bare sediment | 0.43 | 0.22 | 0.27 | Barron et al., 2006 |
|  |  | *P. oceanica* | 7.72 | 3.18 | 16.44 |  |

*The daily values of R and GPP reported in the literature were divided by 24 and 12, respectively, to calculate the hourly values.*

Key drivers of elevated metabolic rates in tropical meadows include greater PAR availability, aboveground biomass, and higher temperatures (Ganguly et al., 2017; Ward et al., 2022). Many tropical species grow near their optimal photosynthetic and physiological conditions (Lee et al., 2007; Koch et al., 2012), efficiently capturing light in shallow, clear waters, which contributes to higher NEP (Ralph et al., 2007). In our study, DO variation corresponds to light intensity (Figs. 3 and 7), suggesting that the elevated GPP observed in seagrass meadows could be driven by higher light intensity. This is likely due to the relatively lower canopy cover of *H. uninervis* and density in SG, which reduces shading within the seagrass. As a result, more light penetrates to the leaves, increasing their photosynthetic surface area and contributing to NEM (Ralph et al., 2007).





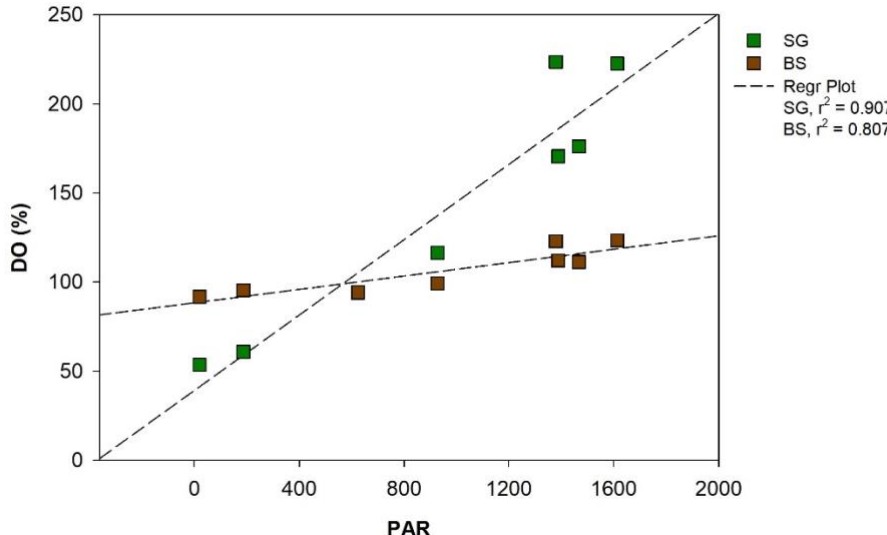

**Figure 7: Regression plot between photosynthetically active radiation (PAR, µmol m$^{-2}$ s$^{-1}$) *vs* dissolved oxygen (DO, %) in restored seagrass (SG, green square) and bare sediment (BS, brown square).**

Several studies indicate that restored seagrass can achieve primary productivity and carbon sequestration levels comparable to natural meadows, although recovery depends on the extent of degradation, restoration success, and site-specific habitat conditions (Oreska et al., 2017; Marbà et al., 2015). For example, long-term research in Florida Bay demonstrated that sediment carbon sequestration rates and plant biomass took nearly a decade to match those of natural meadows (Greiner et al., 2013). The ability of restored meadows to maintain net autotrophy is crucial for their role as carbon sinks (Kennedy et al., 2010). This is particularly relevant for climate change mitigation strategies, where the conservation and rehabilitation of this ecosystem are recognized as natural climate solutions (Griscom et al., 2017). Nonetheless, a recent investigation on restored seagrass exhibits net heterotrophy, as observed by Kindeberg et al. (2024) in both 3-year and 7-year-old meadows in Sweden. A similar pattern also reported in some natural seagrass meadows in Australia (Chen et al., 2019) (Table 1). This discrepancy underscores the variability in seagrass productivity and metabolic processes based on geographical location and





environmental conditions, highlighting the need for region-specific assessments to fully understand seagrass ecosystem dynamics. Long-term studies should also consider temporal and annual variations.

## 4.2 Calcification dynamics in restored seagrass

Our results show that restored seagrass meadows exhibit significantly higher $CaCO_3$ cycling — both formation and dissolution — compared to bare sediments. This corroborates with prior studies, which documented enhanced carbonate dynamics in vegetated habitats relative to unvegetated sediments. For instance, *P. oceanica* and *Thalassia testudinum* meadows have been shown to promote both $CaCO_3$ production and dissolution (Burdige and Zimmerman, 2002; Barrón et al., 2006), with tropical seagrass ecosystems displaying similar patterns (Chou et al., 2021; Fan et al., 2024). Further, our data revealed a typical diurnal pattern, with positive values during daytime (net calcifying) and negative values during nighttime (net dissolving). These findings align with previous estimates, such as those in Florida Bay, which reported similar diurnal calcification dynamics (Yates and Halley, 2006).

The variations of $CaCO_3$ production and dissolution in surface waters and sediment are related to the carbon cycle through photosynthesis and respiration (Yates and Halley 2006). During daylight hours, photosynthesis raises pH and reduces $CO_2$ levels in the water, creating favorable conditions for calcium carbonate precipitation—a process referred to as light–enhanced calcification (Schneider et al., 2009). We found a significant positive correlation between PAR and $n$TA changes ($r^2 = 0.52$, $p<0.05$), suggesting that increased light availability may enhance calcification by photoautotrophs in restored seagrass areas during the day (Fig. 8). Additionally, our data showed a significant negative correlation between $n$TA flux and NEM ($r^2=0.54$, $p<0.01$), indicating that higher photosynthetic activity (positive NEM) promotes calcification by consuming TA, while lower NEM or net heterotrophy contributes to TA production, likely through carbonate dissolution or anaerobic decomposition (Fig. 9). Similar relationships between photosynthesis and calcification have been reported in marine calcifiers (Mallon et al., 2022), and the influence of epiphytic organisms in promoting calcification during active photosynthesis has been highlighted in seagrass meadows such as *P. oceanica* (Barrón et al., 2006). At night, carbonate dissolution predominates as aerobic respiration produces $CO_2$ and carbonic acid in sediment porewater (Eyre et al.,



2014), lowering carbonate saturation and driving mineral dissolution (Burdige and Zimmerman, 2002;
Burdige et al., 2008; Chou et al., 2021; Fan et al., 2024). The degree of dissolution is directly link to the
rate of organic matter decomposition, which depends on the quantity of organic matter, its reactivity, and
oxygen availability (Anderson et al., 2005; Morse et al., 2006). The presence of shoot density and root
biomass in restored seagrass meadows enhances organic matter supply and decomposition in sediment,
further driving nighttime dissolution (Holmer et al., 2013).

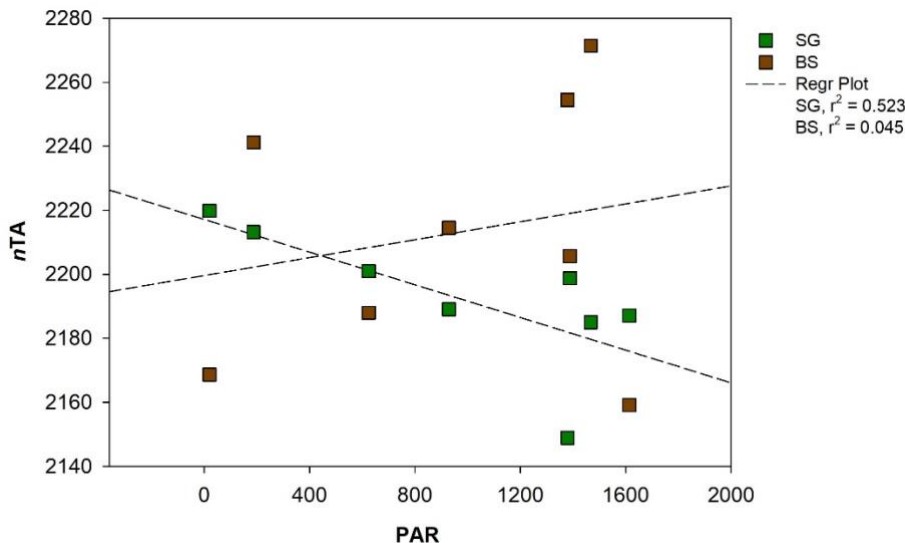


**Figure 8: Regression plot between photosynthetically active radiation (PAR, µmol m$^{-2}$ s$^{-1}$) *vs***
**normalized total alkalinity ($n$TA, µmol kg$^{-1}$) in restored seagrass (SG, green square) and bare**
**sediment (BS, brown square).**



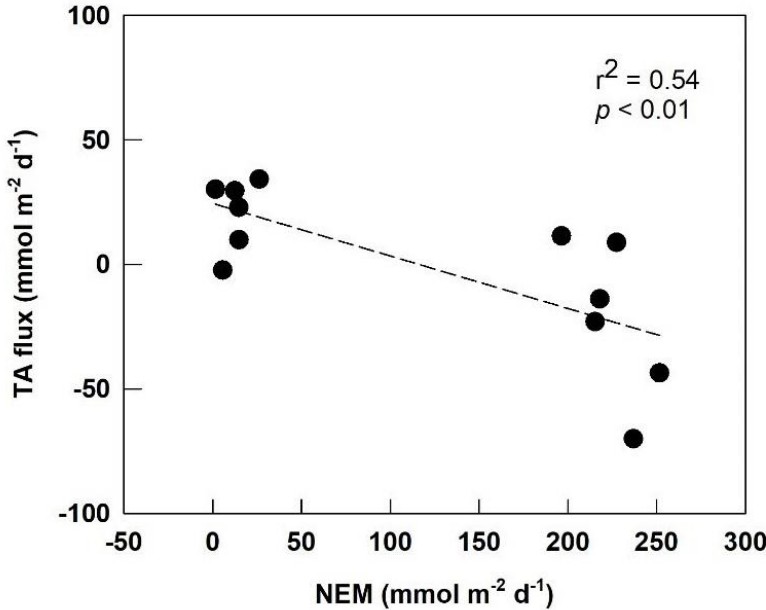

**Figure 9: Linear regression showing the relationship between total alkalinity (TA) flux and net ecosystem metabolism (NEM) in this study.**

Over cumulative days, our NEC measurements indicate that restored seagrass meadows support overall net calcification, whereas BS supports net dissolution. Our estimates are similar to those from Australia (Walker et al., 1988) and seven times higher than Mediterranean seagrass net calcification rates (Barrón et al., 2006), which are 295 g $CaCO_3$ $m^{-2}$ $yr^{-1}$ (8.8 mmol $CaCO_3$ $m^{-2}$ $d^{-1}$) and 51 g $CaCO_3$ $m^{-2}$ $yr^{-1}$ (1.40 mmol $CaCO_3$ $m^{-2}$ $d^{-1}$), respectively. In contrast, our findings are lower than those reported in the Caribbean region of Mexico, where ex situ estimates ranged from 14 to 153 mmol $CaCO_3$ $m^{-2}$ $d^{-1}$ (Enriquez and Schubert, 2014). This highlights the enhanced carbonate production potential in tropical seagrass meadows. A positive net calcification system occurs when $CaCO_3$ precipitation exceeds dissolution within the system (Kleypas et al., 2001; Eyre et al., 2014). Restoration of seagrass meadows provides a substrate for diverse calcifying organisms, including crustose coralline algae, bryozoans, foraminifera, and serpulids, which enhance carbonate production (Beavington-Penney et al., 2005). Epiphytes on seagrass leaves significantly contribute to $CaCO_3$ production, with tropical seagrass





meadows typically supporting higher carbonate loads than temperate ones. Reported production rates
span from 180 g $CaCO_3$ m$^{-2}$ yr$^{-1}$ in Jamaica (Land, 1970) to 2800 g $CaCO_3$ m$^{-2}$ yr$^{-1}$ in Barbados (Patriquin,
1972), underscoring regional variability in seagrass-associated calcification. Moreover, fluctuations in
$CO_3^{2-}$ concentrations are crucial in regulating the capacity of calcifying organisms to form $CaCO_3$. Our
data reveals a consistently higher aragonite saturation state ($\Omega_a$) compared to bare sediment (BS). Notably,
SG environments exhibit significant peaks in aragonite saturation, with a maximum value of 5.686,
whereas the highest $\Omega_{Ar}$ in BS is 3.419. Seagrass photosynthesis raises pH and $\Omega_{Ar}$, enhancing the
calcification of surrounding calcifying organisms (De Beer and Lakrum, 2001). However, the
consumption of TA by calcifiers during the calcification process releases $CO_2$, potentially counteracting
pH increases and partially offsetting the net carbon uptake potential of seagrass ecosystems (Alongi et
al., 2008; Mazarrasa et al., 2015; Saderne et al., 2019). This highlights the dual role of seagrass restoration
in supporting biodiversity and $CO_2$ uptake while influencing carbonate and carbon flux dynamics.
Although the restored seagrass meadow in our study functions as a net calcifying system, TA fluxes
between SG and BS showed no significant difference.

### 4.3 Net carbon uptake of seagrass restoration

In order to estimate the net carbon uptake potential of seagrass restoration, we applied the photosynthesis-
quotient of 1 to calculate $CO_2$ uptake from organic carbon metabolism (Gattuso et al., 1998; Ward et al.,
2022). In terms of carbonate dynamics, each mole of $CaCO_3$ formed releasing 0.6 mmol of $CO_2$ into the
seawater was used (Frankignoulle et al., 1994). The calculated results show that total carbon uptake from
NEM was 208 mmol $CO_2$ m$^2$ d$^{-1}$ in SG and 20 mmol $CO_2$ m$^2$ d$^{-1}$ BS. For NEC, the carbon release in SG
was 6.52 $CO_2$ m$^2$ d$^{-1}$, while for BS, an additional $CO_2$ uptake was -1 mmol $CO_2$ m$^2$ d$^{-1}$. Consequently, the
net carbon uptake is 202 and 21 mmol $CO_2$ m$^2$ d$^{-1}$ for SG and BS, respectively. Our results demonstrate
that the primary productivity of restored seagrass through photosynthesis exceeds the rates of calcification
by 31-fold, suggesting that restored seagrass can act as a net carbon sink. However, further assessments
are necessary to capture temporal variations, as our current measurements are based on daily observations
and one season only.





**4.4 Limitations of ex situ benthic incubation and future research**

We tested the ex situ benthic core incubation approach for restored seagrass meadows, drawing from the existing utilities in some coastal areas and freshwater ecosystems for sulfate and nutrient fluxes (Eyre, et al., 2005, Chen et al., 2019). Overall, the ex situ benthic incubation method provides a significant advantage by measuring both organic and inorganic carbon dynamics simultaneously, addressing a critical gap in previous methods that often overlook carbonate dynamics (Johanssen, 2023). This approach is also useful for assessing seagrass metabolism in subtidal meadows, where collecting data is challenging due to high labor costs and weather conditions. Moreover, some in situ autonomous methods are often expensive and constrained operational periods of only a few weeks due to challenges like sensor error and biofouling (Yates and Halley, 2003; Takeshita et al., 2016). While this approach provides several advantages, one notable limitation is its applicability. Currently, the design is primarily suited for small seagrass, like *H. ovalis*, *H. uninervis*, and *Z. japonica*. It may not be adequate for larger species, like *Enhalus acoroides* and large *Thalassia hemprichii*, due to differences in size and growth characteristics. Moreover, we suggest validating the ex situ results with in situ data to ensure comparability with natural conditions. Future research should integrate ex situ results with in situ data with different geographic and environmental settings to enhance the generalizability of the findings. This will provide a more accurate assessment of seagrass ecosystems' role in global carbon cycling and inform more effective coastal management and conservation practices.

**5 Conclusion**

This study investigates the organic carbon metabolism and carbonate dynamics of replanted SG compared to BS using the ex situ core incubation method. The results show that SG has higher GPP and NEM, while exhibiting similar NEC, making it a stronger carbon sink than BS. The findings highlight the role of seagrass restoration in enhancing carbon removal and contribute to a growing body of literature that highlights the ecological value of restored seagrass meadows. This study represents the first simultaneous quantitative estimate of the effect of both organic carbon metabolism and carbonate dynamics on carbon sequestration of restored seagrass in Southeast Asia, providing valuable insights into the region's carbon



dynamics. We emphasize the need for long–term research on metabolic rates and carbonate dynamics to
account for temporal variations and to fully understand the implications of these processes in carbon
sequestration. This will also help optimize restoration strategies aimed at maximizing carbon sink
potential and mitigating ocean acidification. Furthermore, ex situ benthic incubation proves to be a
valuable tool for assessing carbon fluxes in seagrass meadows, particularly those dominated by pioneering
species, although further in situ assessments are necessary for comprehensive validation.

**Author contribution**

Wen-Chen Chou (WCC) and Jian-Jhih Chen (JJC) conceptualized the research and spearheaded the
implementation. JJC, Mariche B. Natividad (MBN), and Hsin-Yu Chou facilitated sample collection and
analysis. MBN and JJC performed the data analysis, drafted the manuscript, and its revision. WCC and
Lan-Feng Fan reviewed and revised the manuscript. All authors were involved in the finalization of the
manuscript.

**Competing interest**

The authors declare that they have no conflict of interest.

**Data availability**

The datasets in this study will be deposited in DRYAD Data Repository.

**Acknowledgment**

We are grateful to Hsin-Chiao Chang, Yuhann Yokie-Tai, Ping-Chun Chen, and Xin-Yi Wang for the
field sampling and laboratory assistance.

**Financial support**

This work was funded by the National Science and Technology Council of Taiwan under grant numbers
NSTC 113-2119-M-019-008 and NSTC 113-2611-M-019-011, given to WCC.





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
