# Peer review of "Estimation of Metabolic Dynamics of Restored Seagrass Meadows in a Southeast Asia Islet"

_EGUsphere, 2024_

## Author Comment (AC1)

**Author response to RC1**

The authors presented a set of incubation experiments using vegetated and unvegetated sediment cores and explored production, respiration, and carbonate precipitation and dissolution rates for these two type of environments. Their general conclusions are as expected, i.e., seagrass vegetation enhances organic carbonate production along with net calcification, compared with bare sediments.

The manuscript is largely well written and easy to comprehend. The only major comment I have is with the way statistics is presented. It is unclear why the authors chose to report standard error instead of standard deviation for these replicate core incubations. Is the purpose to reduce the size of the error bar? I would report standard deviations instead to show variability.

Response: Thank you for your thoughtful feedback. We initially reported standard error (SE) instead of standard deviation (SD) to emphasize the precision of the mean metabolic estimates for each treatment, as SE reflects how well our sample mean represents the true population mean. However, we understand the importance of representing variability more explicitly. In response to your suggestion, we have revised the figures (Fig. 03 to Fig. 06) to display SD instead of SE, along with the corresponding text revisions.

There are also some confusions with how the two set of incubations are compared, please see detailed comments below.

The graphic abstract is inconsistent with abstract "… resulting in no significant difference in NEC between SG and BS". But in the graphic abstract, NEC = 10.9 and -2.3 in SG and BS sediments.

Response: Thank you for your comment. We acknowledge the concern regarding the apparent discrepancy between the values in the graphic abstract and the conclusion in the text. The values presented in the graphic abstract (NEC = 10.9 for SG and -2.3 for BS) accurately represent the mean NEC fluxes observed in our study. However, statistical analysis using an independent t-test (t = 1.320, df = 10, p = 0.216) indicates no significant difference between the two groups.

The numerical difference in mean NEC values is accompanied by high variability within each group (SG: SD = 15.66, BS: SD = 18.80). This suggests that while a numerical difference exists, the statistical test does not detect a significant effect, likely due to the substantial overlap in variability between the groups. We have provided the statistical results below for your reference.

**Group Statistics**

| | Treatment | N | Mean | Std. Deviation | Std. Error Mean |
|---|---|---|---|---|---|
| NEC | 1.00 | 6 | 10.8704 | 15.66210 | 6.39403 |
| | 2.00 | 6 | -2.3097 | 18.79665 | 7.67370 |

1 – SG; 2 – BS

**Independent Samples Test**

| | | t-test for Equality of Means | | | | | | |
|---|---|---|---|---|---|---|---|---|
| | | | | | | | 95% Confidence Interval of the Difference | |
| | | t | df | Sig. (2-tailed) | Mean Difference | Std. Error Difference | Lower | Upper |
| NEC | Equal variances assumed | 1.320 | 10 | .216 | 13.18009 | 9.98846 | -9.07558 | 35.43576 |
| | Equal variances not assumed | 1.320 | 9.685 | .217 | 13.18009 | 9.98846 | -9.17419 | 35.53437 |

To clarify this in the abstract, we have revised the statement as follows:

*"In contrast, SG exhibited net calcification with positive NEC values (10.9 ± 15.7 mmol CaCO₃ m⁻² d⁻¹), driven by higher daytime carbonate production than nighttime dissolution, while BS showed net dissolution with negative NEC values (-2.3 ± 18.8 mmol CaCO₃ m⁻² d⁻¹). Despite this, high variability in carbonate fluxes led to no significant difference between SG and BS (p>0.05)." (L25-L29)*

Section 2.5, need to spell out assumptions for using short durations (a few hours) to estimate daily rates.

Response: Thank you for your suggestion. We have clarified our assumptions for using short incubation durations to estimate daily rates. Specifically, we used a 12-hour incubation period, consisting of a 6-hour dark incubation to maintain oxygen concentrations above 80% (Eyre et al., 2002) and a 6-hour light incubation to prevent oxygen supersaturation (Olivé et al., 2016). These assumptions have been incorporated in lines 167-169 of the manuscript.

*"We implemented a 6-hour dark incubation period to ensure oxygen concentrations remained above 80% (Eyre et al., 2002) and a 6-hour light incubation period to prevent oxygen from reaching supersaturated levels (Olivé et al., 2016)."*

L102, add citation for Coral Allen Atlas

Response: Thank you for your suggestion. We have added the citation for the Allen Coral Atlas (L105) and included it in the references as follows:

Allen Coral Atlas: Imagery, maps and monitoring of the world's tropical coral reefs. Zendodo. doi.org/10.5281/zenodo.3833242, 2020. (L472-473)

L141, change "checker" to "sonde"

Response: Thank you for your suggestion. We have changed "checker" with "sonde" (L145).

L163, change "difference" to "sum", adding respiration rate and NPP to get GPP.

Response: Thank you for your suggestion. We acknowledge that GPP is often calculated as the sum of NPP and R; however, in our study, we computed hourly GPP as the difference between NPP and R, following Equation 4. This approach maintains consistency with our metabolic flux calculations, where R is expressed as a negative value. Mathematically, subtracting a negative R is equivalent to adding its absolute value, yielding the same result as summing NPP and R. Given this formulation, we believe "difference" accurately reflects our approach. We have also revised our equation 4:

GPP = NPP (positive) – R (negative)          (eq. 4) (L178)

This formulation is consistent with metabolic calculations used in studies such as Chen et al. (2019) and Eyre and Ferguson (2005).

L175, state the duration of alkalinity difference measurements

Response: Thank you for your comment. The duration of alkalinity difference measurements is already stated in the revised manuscript (L187-190).

*"Day and night incubations (lasting 12 hours) were conducted simultaneously with organic carbon metabolism to obtain daily NEC fluxes. The dark period (midnight to dawn) was used to measure*

*nighttime dissolution, while the light period (dawn to noon) was used for daytime calcification. Alkalinity was measured every 3 hours throughout the incubation period.*

L205-207, suggest removing this sentence, otherwise worsening OA at night needs to be included.

Response: Thank you for your suggestion. We have removed the sentence.

L234-235, show the data.

Response: Thank you for your comment. In response, we have included the data as requested.

*"Both R and GPP in SG and BS increased on the second day of incubation [SG (R: -3.1 vs -5.6 mmol $O_2$ $m^{-2}$ $h^{-1}$; GPP: 23.3 vs 24.7 mmol $O_2$ $m^{-2}$ $h^{-1}$); BS (R: -0.4 vs -0.81 mmol $O_2$ $m^{-2}$ $h^{-1}$; GPP: 2.7 vs 3.1 mmol $O_2$ $m^{-2}$ $h^{-1}$)], while NEM in SG (218.04 vs 198.4 mmol $O_2$ $m^{-2}$ $d^{-1}$) and BS (22.3 vs 17.8 mmol $O_2$ $m^{-2}$ $d^{-1}$) showed a slight decrease. However, these changes were not statistically significant."* (L241-245)

Fig. 5, use the same y-axis unit to avoid confusion

Response: Thank you for your feedback. We have revised Figure 5 to ensure consistency in the y-axis units, particularly the scale. However, we have retained the per-hour unit for GPP and R, and per-day for NEM. This distinction is necessary because our incubations only extend until noon, excluding afternoon fluxes, which makes per-hour rates more appropriate for GPP and R. (L246)

[Figure]

**Figure 1: Mean (± SD, standard deviation) values of (a) metabolic rates such as respiration (R), gross primary productivity (GPP), and (b) net ecosystem metabolism (NEM,) of restored seagrass (SG, green bars) and bare sediment (BS, brown bars) in Penghu during the two-day (April 12-13, 2024) incubation (n=9).**

L287, reword to "when GPP is lower".

Response: Thank you for your suggestion. We have revised the sentence to reflect better the findings of Duarte et al. (2010). The revised sentence now reads:

*"According to Duarte et al. (2010), seagrass meadows generally act as autotrophic (NEM > 0) $CO_2$ sinks when GPP exceeds 186 mmol $O_2$ $m^{-2}$ $d^{-1}$, and shift to heterotrophy (NEM < 0) when GPP falls below this threshold."* (L295-297)

Table 1, why not use the same unit to facilitate comparisons?

Response: Thank you for your comment. We chose to use per-hour units to facilitate more accurate comparisons, as our incubation periods differ: respiration was measured from 12:00 midnight to 6:00 AM, and NPP from 6:00 AM to 12:00 noon. Using per-hour units ensures consistency within these time frames and avoids assumptions about afternoon fluxes.

Fig. 7, need error bars

Response: We thank the reviewer for their suggestion. We have added error bars to Figure 7, as well as Figure 8 and 9. Please see the revised figures below.

[Figure]

Figure 7: Regression plot between photosynthetically active radiation (PAR, μmol m-2 s-1) vs dissolved oxygen (DO, %) in restored seagrass (SG, green square) and bare sediment (BS, brown square). Error bars represent standard deviation (SD).

[Figure]

Figure 8: Regression plot between photosynthetically active radiation (PAR, μmol m-2 s-1) vs normalized total alkalinity (nTA, μmol kg-1) in restored seagrass (SG, green square) and bare sediment (BS, brown square). Error bars represent standard deviation (SD).

[Figure]

Figure 9. Linear regression showing the relationship between total alkalinity (TA; mmole m$^{-2}$ d$^{-1}$) flux and net ecosystem metabolism (NEM; mmol m$^{-2}$ d$^{-1}$) in restored seagrass meadows. Error bars represent standard deviation (SD).

L356, remove "shoot density" or change to high shoot density and root biomass.

Response: Thank you for your suggestion. We have revised the text accordingly by changing it to 'high shoot density and root biomass' as recommended.

"High shoot density and root biomass in restored seagrass meadows enhance organic matter supply and decomposition in sediment, further driving nighttime dissolution (Holmer et al., 2013)." (L367-L369)

L383, but earlier in the text (L216), Ω between the two sets are not significantly different, which contradict with L213 however.

Response: Thank you for your helpful comment. We understand the concern regarding the apparent contradiction between the statements in L383 and L216, and we appreciate the opportunity to clarify this.

In L216, we reported that there was no significant difference in ΩAr between SG and BS (p = 0.511) based on our Mann–Whitney test. This statistical result indicates no significant distinction in aragonite saturation state between the two environments.

In the Discussion (L383), we initially stated that SG environments exhibit significantly higher aragonite saturation than BS, with notable peaks in SG. To better align with the statistical results, we have revised the discussion to focus on the average ΩAr values rather than the maximum values. These changes are reflected in Lines 396-397.

*"Our data reveal a higher mean $\Omega_{Ar}$ in SG (3.14 ± 1) compared to BS (2.72 ± 0.4)."*

L391-392, with NEC much different, why alkalinity fluxes are similar? Or is it because the variations are larger than the difference of the means?

Response: Thank you for your comment. Despite the apparent differences in NEC between SG (10.9 ± 15.66) and BS (-2.3 ± 18.80), the similarity in alkalinity fluxes is primarily attributed to the high variability within each group. While the mean NEC values for SG and BS differ, with SG showing a positive value and BS showing a negative value, the considerable overlap in their standard deviations and wide confidence intervals (SG: -5.57 to 27.31, BS: -22.04 to 17.42) suggest that the variability within each group outweighs the difference in their averages. The differences in NEC, therefore, may not be statistically significant, despite the numerical distinction in the means.

L396-397, note the cited study use a seawater that may or may not be the same as the seawater in your case, so it is useful to do some calculation.

Response: Thank you for your suggestion. We appreciate the note regarding the seawater conditions used in the cited study by Frankignouelle (1994). We acknowledge the importance of considering potential differences in environmental factors, such as seawater chemistry, that could affect carbonate dynamics. To address this, we have calculated the size of $CO_2$ source or sink ($\Phi$) values using the equation described by Humphreys et al. (2018), based on the specific seawater parameters observed in our study. Our calculations yielded a $\Phi$ value of 0.61 for the SG system and 0.65 for the BS system.

In the revised manuscript, we state:

"In terms of carbonate dynamics, we applied $\Phi$, as described by Humphreys et al. (2018), to calculate the size of $CO_2$ source or sink for each system. In the SG system, which is net calcifying, $\Phi$ indicates a $CO_2$ source, with 0.61 moles of $CO_2$ released into the seawater for each mole of $CaCO_3$ precipitated. In contrast, the BS system, which is net dissolving, $\Phi$ represents a $CO_2$ sink, with 0.65 moles of $CO_2$ absorbed for each mole of $CaCO_3$ dissolved. These values are comparable to previous findings, which reported a $CO_2$ flux-to-$CaCO_3$ precipitation ratio of 0.63 (Frankignoulle et al., 1994; Smith, 2013; Mazarrasa et al., 2015)." (L408-414)

---

## Author Comment (AC2)

**Author response to RC2**

Natividad and others explore the carbon balance and metabolism of a restored seagrass meadow. The analysis is largely interesting and methods sound, but a number of aspects of the presentation need to be improved and I have concerns about how the in situ results are translated to the real world environment.

Response: We sincerely thank the reviewer for the thorough and constructive evaluation of our manuscript "Estimation of Metabolic of Restored Seagrass Meadows in a Southeast Asia Islet Insights from Ex Situ Benthic incubation.

We acknowledge the reviewer's concern regarding the translation of ex situ results to real-world in situ conditions. In this study, we used ex situ benthic incubation as a practical and widely used method to quantify seagrass metabolism, especially in subtidal systems where in situ measurements are logistically challenging. This method has been used in several studies and provide successful and valuable dataset. While we recognize that ex situ conditions may differ from natural underwater environments, we carefully designed our setup to mimic field conditions.

We have now revised the manuscript to clarify this methodological approach and added a statement discussing its strengths and limitations, including the need for future validation with in situ data under varying environmental conditions.

Line 122-125 "*This method offers a feasible approach for quantifying seagrass metabolism, especially in subtidal systems where in situ measurements are often logistically challenging. While ex situ conditions may differ from natural underwater environments, we carefully designed our setup to closely replicate field conditions, including natural light exposure and ambient temperature, to ensure ecological relevance.*"

Graphical abstract: units are needed on some terms (namely calcification).

Response: Thank you for the comment. We have revised the graphical abstract to include appropriate units ("mmol"), and the calcification rates are now clearly labeled as $\text{mmol m}^{-2}\,\text{h}^{-1}$. (Line 38)

During the revision process, we also identified an error in the previously reported values for calcification and dissolution. Specifically, although the units were indicated as $\text{mmol m}^{-2}\,\text{h}^{-1}$, the values reflected daily rates. We have now converted and corrected these values to accurately report hourly rates ($\text{mmol m}^{-2}\,\text{h}^{-1}$), as originally intended. These changes do not affect the overall interpretation or conclusions of the study.

[Figure]

41: are there 72 species globally? Are species-level differences relevant here? Does this apply to just the grasses?

Response: Yes, there are approximately 72 seagrass species globally (Fourqurean et al, 2012; Short et al., 2011). Species-level differences are relevant, particularly in studies of productivity and metabolism, as seagrass species vary in morphology, physiology, and ecological function. These differences can influence carbon and carbonate dynamics, and thus are important to consider in site-specific assessments. To clarify, the term "seagrasses" here specifically refers to marine angiosperms, which are taxonomically distinct from terrestrial grasses.

145: did this reflect the PAR making it to the grasses in the natural environment with a larger water column on the order of 2-4 m? (see also line 197 and elsewhere). This feeds into the explanation on line 305: is PAR actually higher at leaf level in situ?

Response: We thank the reviewer for this insightful comment. In our study, PAR sensors were deployed in air to record incident solar radiation. However, based on our laboratory tests, when sediment cores were transferred to the incubation tanks, the light intensity decreased by approximately 50% due to attenuation through the water column and incubation setup. This attenuation resulted in light levels within the incubation tanks that were comparable to those in the natural seagrass meadows where samples were collected.

While we acknowledge that some differences remain between ex-situ and in-situ light environments. We believe that our approach provides a reasonable approximation of field conditions in the absence of in-situ incubations. Indeed, previous research has shown that ex-situ and in-situ incubations can yield comparable metabolic estimates, supporting the validity of our approach (Maher and Eyre 2011). We have added this clarification to the limitation of ex situ incubation and future research section to acknowledge this limitation and highlight the value of future in situ incubations for more accurately capturing the light environment experienced by seagrass leaves in their natural habitat.

Line 451-457

*"Moreover, we suggest validating the ex situ results with in situ data to ensure comparability with natural conditions, particularly the effects of light attenuation. Our measurements were obtained under ex-situ conditions in a shallow water column, which likely exposed the cores to higher irradiance than would be encountered in situ at different seagrass depths (2–4 m). While previous research has shown that ex situ and in situ incubations can yield comparable metabolic estimates, supporting the validity of our approach (Maher and Eyre, 2011), we acknowledge the need for future in situ incubations to more accurately capture the natural light environment experienced by seagrass leaves."*

Ref: Maher, D., & Eyre, B. D. (2011). Benthic carbon metabolism in southeast Australian estuaries: Habitat importance, driving forces, and application of artificial neural network models. Marine Ecology Progress Series, 439, 97-115. (Line 637-639)

a bit critical of using hours for measurements because this is not an SI unit. Aggregating to days is a different story because this is an aggregation. I know that the community often uses hours, but a lot can happen in an hour. Perhaps note that these units are to compare against other studies.

Response: We acknowledge the reviewer's point that hours are not SI units. However, regarding the temporal resolution of measurements, hourly measurements are standard practice in marine and plant metabolism studies because they capture fine-scale variations in light availability that are critical to photosynthetic processes.

We agree that substantial changes can occur within an hour; therefore, we reported both hourly and daily rates in the main text. Hourly rates were used to examine diel variations in metabolic processes between bare and seagrass habitats, while daily rates were presented as net values to provide an integrated perspective on the overall carbon dynamics. This approach captures detailed variability across time scales (from hourly to daily). Moreover, several previous studies have reported data in a similar manner, facilitating comparisons across different systems and contributing to broader synthesis efforts. To clarify this, we have added the following statement to the method section on Benthic flux rates calculation Line 196-198: "*In this study, both hourly and daily rates were reported. Hourly rates allow us to examine diel variations in metabolic processes, while daily rates provide an integrated view of overall carbon dynamics, facilitating comparison with existing literature.*"

Fig. 3: does this pass a colorblindness check? At a minimum use differently shaped symbols. Also Fig. 4. And especially Figure 8. This figure would not be interpretable if printed in black and white, and not everyone has a color printer.

Response: Thank you for the helpful suggestion. We have revised Figure 3 (Line 230), Figure 4 (Line 2234), Figure 7 (Line 335), and Figure 8 (Line 386) to incorporate distinct shapes (e.g., circles and triangles) along with color, ensuring they are accessible for colorblind readers and interpretable in black-and-white printing.

[Figure]

**Figure 1:** Diurnal pattern of dissolved oxygen (DO, a) in replanted seagrass (SG, green triangle) and bare sediment (BS, brown circle) (n=9, mean ± SD), and photosynthetically active radiation (PAR, b) during the two-day (April 12-13, 2024) incubation.

[Figure]

**Figure 2: Total scale pH (pH$_T$, a), normalized dissolved inorganic carbon (*n*DIC, b), normalized total alkalinity (*n*TA, c), partial pressure of carbon dioxide (*p*CO$_2$, d), and aragonite saturation state (Ω$A_R$, e) in replanted seagrass (SG, green triangle) and bare sediment (BS, brown circle) during the two-day (April 12-13, 2024) incubation. n=3, mean ± SD.**

[Figure]

**Figure 3: Regression plot between photosynthetically active radiation (PAR, µmol m$^{-2}$ s$^{-1}$) *vs* dissolved oxygen (DO, %) in restored seagrass (SG, green triangle) and bare sediment (BS, brown circle). Error bars represent standard deviation (SD).**

[Figure]

**Figure 4: Regression plot between photosynthetically active radiation (PAR, µmol m$^{-2}$ s$^{-1}$) *vs* normalized total alkalinity (*n*TA, µmol kg$^{-1}$) in restored seagrass (SG, green triangle) and bare sediment (BS, brown circle). Error bars represent standard deviation (SD).**

[Figure]

**Figure 5: Linear regression showing the relationship between total alkalinity (TA, mmol m⁻² d⁻¹) flux and net ecosystem metabolism (NEM, mmol m⁻² d⁻¹) in restored seagrass meadows and bare sediment. Error bars represent standard deviation (SD).**

From the discussion, why is R suppressed in the seagrass ecosystem? The full mechanisms might be clear but it's important to explain what is happening to the best that the data will allow.

Response: We appreciate the reviewer's observation and agree that further explanation of the suppressed respiration (R) in the seagrass (SG) area is important, even if the full mechanisms cannot be entirely resolved with the current dataset. In response, we have revised the discussion to elaborate on the most likely factors contributing to the lower R values observed.

Specifically, we added that the SG beds are located in carbonate-rich sediments, which typically contain lower organic matter than siliciclastic or muddy sediments (Belshe et al., 2018; Kindeberg et al., 2018). This limits the availability of labile substrates for microbial decomposition. Furthermore, the organic matter derived from seagrass detritus is generally more refractory and less labile, which reduces its accessibility for microbial breakdown and thus suppresses heterotrophic respiration. While seagrasses can transport oxygen to belowground tissues via internal aerenchyma (Borum et al., 2006), supporting aerobic respiration, the combination of low organic content and substrate quality appears to constrain microbial activity and oxygen consumption.

We have incorporated this explanation into the revised manuscript (Lines 324-333), clarifying the likely mechanisms of respiration rates.

---

## Author Response (AR2)

Dear Dr. Stoy,

Thank you for your positive evaluation of our revised manuscript and for acknowledging the improvements made. We sincerely appreciate your careful attention to detail and are pleased to address the remaining minor comments as follows:

**1. Abstract unnecessarily wordy**

*Response:*
We thank the reviewer for this valuable comment. We have revised the abstract to enhance clarity and conciseness. Redundant words and less essential details were removed to better highlight the objectives, key findings, and significance of the study. As a result, the abstract has been shortened from 303 to 221 words.

**2. Didn't address question in the manuscript regarding the 72 species. What does this refer to? Is it important?**

*Response:*
We thank the reviewer for raising this point again and apologize for not fully addressing it in the previous revision. In the revised Introduction, we now clarify that seagrass meadows are marine angiosperms comprising approximately 72 species globally (Short et al., 2011). Despite their limited taxonomic diversity, these species exhibit important differences in traits such as morphology and productivity, which can influence their carbon storage potential. This clarification is now reflected in the revised text:

*"Seagrass meadows are marine angiosperms comprising approximately 72 species globally (Short et al., 2011). Although they occupy just 0.1% of the ocean's surface and have limited taxonomic diversity, they are highly productive and ecologically significant ecosystems in the marine environments (Fourqurean et al., 2012; Short et al., 2011). L39-40*

*"Nevertheless, carbon storage capacity can vary depending on species-specific traits, geographical location, and environmental conditions (Duarte et al., 2010; Fourqurean et al., 2012)." L48-L49*

**3. Line 126 – "Let the reader decide if it's feasible."**

*Response:*
We appreciate this suggestion and have revised the sentence to present the methodological approach more objectively. The evaluative term "feasible" has been removed. The sentence now reads:

*"This method was used to quantify seagrass metabolism, particularly in subtidal systems where in situ measurements are often logistically challenging." L119-L120*

**Additional Editorial Revisions:**

We also thank Ms. Katja Gänger for reviewing the uploaded files. In response to the notification:

a) Page numbers have now been added to the first two pages of the manuscript.

b) The "Author Contributions" section has been updated to use only the initials of the authors' names, in accordance with the formatting requirements.

Sincerely yours,

**Wen-Chen Chou**
On behalf of all authors